# Targeted Interventional Therapies for the Management of Postamputation Pain: A Comprehensive Review

**DOI:** 10.3390/biomedicines13071575

**Published:** 2025-06-27

**Authors:** Dunja Savicevic, Jovana Grupkovic, Uros Dabetic, Dejan Aleksandric, Nikola Bogosavljevic, Uros Novakovic, Ljubica Spasic, Slavisa Zagorac

**Affiliations:** 1Special Hospital for Rehabilitation and Orthopedic Prosthetics, 11000 Belgrade, Serbia; ljubica.spasic@gmail.com; 2Clinic for Orthopedic Surgery and Traumatology, University Clinical Center of Serbia, 11000 Belgrade, Serbia; grupkovicjovana@gmail.com (J.G.); urosdabetic1983@gmail.com (U.D.); gmcuros@gmail.com (U.N.); slavisa.zagorac@gmail.com (S.Z.); 3Faculty of Medicine, University of Belgrade, 11000 Belgrade, Serbia; boga19@gmail.com; 4Institute for Orthopedic Surgery “Banjica”, 11000 Belgrade, Serbia; aleksandricdejan@gmail.com

**Keywords:** postamputation pain, phantom limb pain, residual limb pain, neuromodulation, spinal cord stimulation, dorsal root ganglion stimulation, peripheral nerve stimulation, cryoneurolysis, radiofrequency ablation

## Abstract

Postamputation pain (PAP), including residual limb pain (RLP) and phantom limb pain (PLP), remains a significant and debilitating complication after limb loss. Despite advances in pharmacological management, many patients experience inadequate pain relief, underscoring the need for alternative therapeutic strategies. **Objective:** This narrative review critically synthesizes current interventional therapies for PAP, focusing on mechanisms, clinical efficacy and practical application. **Methods:** A literature search was conducted in PubMed, EMBASE, Scopus and Web of Science databases for studies published between 2015 and 2025. Relevant articles on peripheral nerve interventions as well as different neuromodulation techniques were included. **Results:** Peripheral interventions (such as alcohol neurolysis, radiofrequency ablation (RFA) and cryoneurolysis (CNL)) and neuromodulation techniques (including spinal cord stimulation (SCS), dorsal root ganglion (DRG) stimulation and cauda equina stimulation (CES)) demonstrate promising outcomes for PAP. Peripheral nerve stimulation (PNS) shows favorable safety and efficacy profiles and may help prevent the chronification of pain. **Conclusions:** Contemporary interventional therapies represent valuable options in the multidisciplinary management of PAP. Nevertheless, further research is required to standardize clinical algorithms, optimize therapeutic decision-making and improve long-term outcomes and quality of life for individuals with PAP.

## 1. Introduction

Postamputation pain (PAP) is a complex and debilitating condition that significantly compromises the quality of life in individuals after limb loss. It comprises two distinct but often overlapping entities: residual limb pain (RLP) and phantom limb pain (PLP). RLP is a frequent long-term complication following amputation and is strongly associated with reduced quality of life. PLP affects the majority of individuals within the first months postamputation, with prevalence increasing over time. Major limb amputees, particularly those with lower extremity loss, experience significant declines in physical functioning and overall life expectancy. Research has shown higher rates of psychopathology and mortality among individuals suffering from chronic PAP compared to those with similar injuries but without persistent pain. RLP and PLP are underpinned by different, yet sometimes overlapping, pathophysiological mechanisms [1,2]. RLP is characterized by nociceptive or neuropathic pain localized within the residual limb, often attributed to peripheral processes such as neuroma formation, prosthetic complications or local musculoskeletal changes [2]. Conversely, PLP involves the perception of pain in the absent, amputated limb and is associated with maladaptive central nervous system plasticity, including cortical reorganization and altered somatosensory processing. Systematic analyses estimated the prevalence of PAP to be approximately 64% among individuals with amputations—the prevalence was significantly higher in developed countries (66.6%) compared to developing countries (54%), potentially due to differences in healthcare infrastructure, surgical standards and patient follow-up—and the prevalence of RLP to be up to 50% of amputees shortly after surgery and persisting in approximately a quarter of individuals even years following amputation. Factors such as the etiology of the amputation, the level of amputation and the presence of pre-existing pain syndromes are important determinants of pain severity and chronicity, with malignancy-associated and upper limb amputations often associated with more severe pain profiles [1,2,3]. Therapeutic strategies for PAP encompass pharmacological treatments (e.g., gabapentinoids, antidepressants and NMDA antagonists), injections and percutaneous procedures (e.g., local anesthetics, botulinum toxin and sympathetic blocks), surgical interventions (e.g., targeted muscle reinnervation, regenerative peripheral nerve interface and neuroma excision), complementary and integrative techniques (e.g., mirror therapy, hypnosis and neurofeedback), as well as assistive technologies (e.g., transcutaneous electrical nerve stimulation, virtual reality and advanced prosthetic interfaces). Despite the widespread use of pharmacological strategies—including opioids, antidepressants, anticonvulsants and N-methyl-D-aspartate receptor antagonists—the complex nature of PAP often necessitates a more comprehensive approach. This highlights the importance of integrating pharmacological with interventional and minimally invasive techniques to achieve optimal and sustainable pain control [4,5].

Interventional therapies, such as ultrasound-guided alcohol neurolysis (UGANL), radiofrequency ablation (RFA), cryoneurolysis (CNL), spinal cord stimulation (SCS), radiofrequency of the dorsal root ganglion (DRG), cauda equina stimulation (CES) and peripheral nerve stimulation (PNS) offer targeted modulation of pain pathways and represent promising modalities for improving outcomes in this population [1,5,6,7].

Given the multifactorial and heterogeneous nature of PAP and the expanding array of available interventional treatments, this comprehensive narrative review aims to critically synthesize current evidence regarding targeted interventional strategies for the management of PAP. Emphasis will be placed on the pathophysiological principles, clinical efficacy and practical application of these techniques, providing a framework for optimizing care in individuals with PAP.

## 2. Methodology

This narrative review was conducted to provide a comprehensive and critical overview of contemporary interventional strategies for the management of PAP. Relevant articles were identified through searches of the PubMed, EMBASE, Scopus and Web of Science databases, focusing on studies published between January 2015 and March 2025. After screening, a total of 27 studies were included for detailed analysis. The search terms included combinations of the following keywords: (“postamputation pain” OR “phantom limb pain” OR “residual limb pain”) AND (“neuromodulation” OR “spinal cord stimulation” OR “dorsal root ganglion stimulation” OR “peripheral nerve stimulation” OR “cryoneurolysis” OR “radiofrequency ablation”). Manual searches of the reference lists of key articles were also performed to identify additional relevant studies.

Inclusion criteria were as follows:

Peer-reviewed articles published in English;

Studies evaluating interventional strategies (acute pain treatment and surgical methods were excluded unless part of combined approaches);

Original research articles, systematic reviews and narrative reviews published in peer-reviewed journals addressing PAP management.

Exclusion criteria included the following:

Studies focusing solely on surgical management or perioperative pain management;

Case reports without procedural detail or outcome reporting;

Abstracts, conference proceedings, letters to the editor and editorial comments.

Given the heterogeneity of available evidence, including heterogeneity in study design, patient populations and reported outcomes, a narrative review approach was selected and a formal systematic review protocol (e.g., PRISM guidelines) was not applied. Instead, a comprehensive but flexible approach was used to identify and critically appraise the available literature relevant to contemporary interventional management of PAP.

## 3. Pathophysiology of Postamputation Pain

PAP is a complex clinical entity resulting from multifactorial pathophysiological processes in both the peripheral and central nervous system. PAP encompasses RLP and PLP, each with distinct but overlapping underlying alterations [1,8].

### 3.1. Nociceptive Mechanisms

Nociceptive components of PAP are predominantly associated with RLP. Following amputation, tissue and nerve injury lead to the release of proinflammatory cytokines and recruitment of immune cells, contributing to peripheral sensitization. These processes activate nociceptors in the residual limb, resulting in mechanical hyperalgesia and spontaneous pain at the amputation site. Local neuroma formation can further exacerbate nociceptive input through disorganized axonal sprouting and persistent activation of peripheral pain fibers [8,9,10].

### 3.2. Neuropathic Mechanisms

The neuropathic aspect of PAP arises from damage to the peripheral nervous system and subsequent changes in the dorsal root ganglia and spinal cord. Following axotomy, ectopic discharges from hyperexcitable dorsal root ganglia neurons—driven by upregulation of voltage-gated sodium channels—result in spontaneous pain sensations independent of external stimuli. These aberrant signals trigger central sensitization, a process mediated by increased activity of N-methyl-D-aspartate receptors in the dorsal horn, which reduces the nociceptive threshold and facilitates recruitment of non-nociceptive afferents into pain circuits. In addition, brain-derived neurotrophic factor released from activated microglia enhances N-methyl-D-aspartate receptor activity, creating a positive feedback loop that sustains spinal hyperexcitability. These changes explain the persistence of symptoms such as allodynia and hyperalgesia in individuals with chronic neuropathic pain after amputation [8,9,10].

### 3.3. Nociplastic and Cortical Contributors

Nociplastic pain in PAP is reflected in the functional and structural reorganization of the central nervous system. Amputation induces remapping of the somatosensory and motor cortices, with neighboring cortical representations encroaching upon the deafferented limb areas. The degree of this cortical reorganization has been positively correlated with the intensity of PLP [9,10].

Understanding the interplay between peripheral and central mechanisms is critical for the development of targeted interventions. Given the heterogeneity of mechanisms underlying RLP and PLP, individualized multimodal therapeutic approaches are essential for optimizing pain control in patients with PAP.

## 4. Contemporary Interventional Strategies for the Management of PAP

### 4.1. Introduction to Interventional Therapies

PAP remains a complex therapeutic challenge due to its heterogeneous pathophysiology, involving both peripheral and central mechanisms. While pharmacological and conservative measures often represent the first line of management, a substantial proportion of patients fail to achieve satisfactory pain relief. In these cases, the therapeutic focus could shift toward interventional approaches, aiming to target specific pain generators more precisely.

Contemporary interventional strategies encompass a broad range of procedures directed at different anatomical and neurophysiological structures involved in PAP. These approaches include peripheral nerve-targeted interventions (such as UGANL, RFA and CNL) and neuromodulation techniques (including SCS, DRG stimulation, CES and PNS). Understanding the indications, mechanisms, advantages and limitations of each interventional modality is crucial for optimizing clinical outcomes and tailoring therapy to the individual patients’ needs.

The following sections provide a detailed overview of contemporary interventional options and the available clinical evidence supporting their efficacy in the management of PAP.

### 4.2. Peripheral Nerve Interventions

Peripheral nerve-targeted interventions represent a valuable option within the interventional management of PAP, particularly in patients with RLP. Techniques such as UGANL, RFA and CNL offer targeted, minimally invasive options to disrupt pain transmission pathways at the site of pathology.

#### 4.2.1. Ultrasound-Guided Alcohol Neurolysis

UGANL involves the percutaneous injection of ethanol which induces chemical destruction of sensory fibers, providing pain relief in cases of refractory residual limb pain caused by neuromas, scar entrapment or other localized peripheral mechanisms. Zhang et al. demonstrated that ultrasound guidance allows precise targeting of the neuroma, increasing efficacy while minimizing complications. Approximately 54% of patients achieved pain relief following 1–3 alcohol neurolysis treatments, with sustained benefits up to six months. This technique could be considered as useful intervention for localized neuroma-related pain before proceeding to more invasive modalities [11].

#### 4.2.2. Radiofrequency Ablation

RFA reduces pain transmission by applying controlled thermal injury to nociceptive fibers, targeting not only neuromas but also aberrant neural activity arising from other peripheral pain generators in the stump. Zhang et al. reported that RFA could be an effective alternative procedure to UGAL for the patients who did not experience satisfactory outcomes with alcohol injections. RFA thus serves as an important step in the interventional algorithm for painful residual limb neuromas [11]. A pilot study by Pu et al. explored the impact of ultrasound-guided RFA on 18 individuals suffering from painful stump neuromas. The findings confirmed a marked and lasting reduction in both residual limb and phantom limb pain, without the occurrence of serious adverse events. Importantly, the treatment response appeared independent of demographic factors such as age and sex or clinical variables like symptom duration. One case report within the study also described improvement in phantom limb sensation post-treatment, indicating that RFA might exert broader effects on altered sensory perception in amputees [12].

These findings confirm that ultrasonography-guided RFA represents a safe, minimally invasive and radiation-free technique that can provide long-term relief of both RLP and PLP in patients [11,12].

#### 4.2.3. Cryoneurolysis

CNL is a minimally invasive technique that targets peripheral nerves through the application of extremely low temperatures, leading to a reversible disruption of nerve conduction. It disrupts peripheral nociceptive conduction through cold-induced Wallerian degeneration, reducing ectopic activity in neuromas as well as afferent input from other pain-generating structures within the residual limb. The structural integrity of axons is often preserved, allowing for subsequent nerve regeneration. Although promising and supported by evidence in various chronic pain conditions, results in PLP have been variable. Ilfeld et al. reported in a randomized controlled trial comparing ultrasound-guided cryoneurolysis to sham intervention found no significant benefits at 4 months. However, post hoc analysis suggested that treatment response may vary depending on the level of amputation, with more favorable outcomes observed among patients with transtibial amputations. These subgroup differences imply that anatomical factors, including nerve accessibility and treatment site, could influence efficacy. These findings highlight the potential of CNL in specific clinical scenarios but also underscore the need for further research to optimize protocols and better define its role in PLP management [13,14].

Peripheral nerve interventions are summarized in Table 1.

### 4.3. Neuromodulation Strategies

Neuromodulation has emerged as a cornerstone in the interventional management of PAP, especially for patients with PLP or refractory RLP. By directly targeting neural circuits implicated in the maintenance of chronic pain, neuromodulation offers a non-pharmacological avenue to achieve pain relief in patients who have previously tried and failed less invasive therapeutic modalities.

#### 4.3.1. Spinal Cord Stimulation

SCS remains one of the most extensively studied neuromodulation techniques for the treatment of PAP. SCS transmits persistent electrical signals to the dorsal columns of the spinal cord through the epidural or subdural placement and modulates pain signaling pathways. SCS modulates dorsal column activity and engages descending inhibitory pathways, reducing the central amplification of pain signals characteristic of PLP. Systematic reviews and case series have demonstrated that SCS can provide significant pain relief in a subset of patients with PLP [15,16,17].

Corbett et al. reported that although evidence is limited primarily to small, uncontrolled studies, some patients may achieve meaningful pain reduction following SCS; sustained benefits were observed in a much smaller proportion. Most included studies were small and uncontrolled, with significant methodological limitations [15].

Also, Jaffee et al. analyzed previously published reports on the use of SCS for PLP and included 33 patients over five studies that met inclusion criteria. Most cases involved epidural lead placement and reported complications were minimal. They identified improvements by at least 50% in pain reduction, reinforcing the role of SCS as a viable option for patients with intractable PLP [16].

Aiyer et al. systematically reviewed 12 studies investigating SCS for the treatment of PLP. Of these, seven studies reported clinically meaningful pain relief, while five demonstrated minimal or no benefit. Pain outcomes were assessed using both objective tools, such as the Visual Analog Scale and the McGill Pain Questionnaire, and subjective patient-reported measures. Overall, the findings suggest that SCS may provide relief in select patients, particularly when the pain is neuropathic in origin in contrast to non-neuropathic conditions like poorly fitting prostheses or local stump infections., but its efficacy remains inconsistent across the broader PLP population [18].

Although current evidence indicates that SCS may offer meaningful relief in certain cases of PLP, its precise role and long-term efficacy remain to be fully established.

#### 4.3.2. Radiofrequency Stimulation of the Dorsal Root Ganglion

DRG stimulation represents a more selective form of neuromodulation, capable of providing focused coverage of targeted areas, and it offers highly focal modulation of sensory input [6,19].

Hunter et al. introduced the concept of using selective radiofrequency stimulation of the DRG to predict optimal neuromodulation targets prior to permanent implantation. In their case series, patients achieved 60–90% pain reduction by targeting specific dorsal root ganglion levels identified through preoperative mapping [19].

Eldabe et al. retrospectively analyzed outcomes in eight patients with phantom limb pain that underwent DRG neuromodulation after successful trial stimulation. Most had previously failed conventional treatments. After a mean follow-up of 9 (±6.3) months, five patients reported meaningful pain relief, including one with complete resolution. The average pain reduction was 52%, and no complications were reported. In three cases, pain relief diminished over time, partly due to suboptimal lead placement. Paresthesias were well localized to the stump and in some patients extended into the phantom limb [20,21].

#### 4.3.3. Spinal Cauda Equina Stimulation

CES is a novel and less commonly utilized approach that may be particularly useful in cases where traditional SCS fails to adequately cover the painful region.

Lee et al. reported a case on CES for PLP, by inserting the needle through L5-S1 interlaminar space with the tip of the needle being placed at the upper L4 vertebral body. It resulted in significant (80%) and sustained pain relief in a patient with refractory PLP. The authors emphasized that CES allows for more distal targeting of nerve roots, potentially offering superior coverage for distal extremity pain compared to conventional SCS. Although further research is warranted, CES represents a potentially innovative alternative for selected patients with challenging PLP presentations [7].

### 4.4. Peripheral Neuromodulation: Peripheral Nerve Stimulation

PNS has emerged as an important modality in the interventional management of PAP, leveraging both peripheral and central mechanisms to achieve analgesic effects. By directly targeting peripheral nerves, PNS provides focused neuromodulation with the potential to disrupt nociceptive signaling and promote central neural reorganization. Several studies have investigated the efficacy of PNS following limb amputation [22,23,24].

Gilmore et al. reported in a multicenter, randomized, placebo-controlled trial evaluating PNS in lower limb amputees with chronic neuropathic pain. In the initial phase, participants receiving 4 weeks of active PNS showed significantly higher rates of ≥ 50% pain reduction compared to placebo. Additionally, after 8 weeks of therapy (including crossover), 67% of patients in the PNS group reported clinically meaningful pain reduction, with 80% reporting decreased pain interference [25]. In an extended 12-month follow-up of a subgroup, 67% of participants continued to report ≥ 50% pain relief, along with reductions in pain interference and depressive symptoms, supporting the durability of PNS effects in managing PAP [26].

Cohen et al. evaluated percutaneous PNS in patients with chronic PAP and found significant improvements in pain scores, functional outcomes and patient satisfaction. Across the included data, approximately 75% of patients achieved clinically meaningful pain relief (≥ 50%) following up to 60 days of therapy. These reductions in pain were often accompanied by improvements in functional capacity and reduced pain interference in daily life. Multiple studies which they evaluated emphasized the minimally invasive nature of PNS and its potential to fill the therapeutic gap between conservative management and more invasive neuromodulation techniques [27].

Although high-quality, large-scale randomized trials are still limited, the cumulative evidence suggests that PNS is a relatively safe and effective intervention for managing both acute and chronic phases of PAP. Given its favorable safety profile and growing body of supporting evidence, PNS represents a valuable addition to the multimodal interventional strategies available for patients with PAP.

Neuromodulation strategies are summarized in Table 2.

## 5. Discussion

### 5.1. Critical Appraisal of Contemporary Interventional Strategies

The management of PAP remains a significant clinical challenge due to the complex interplay of peripheral and central mechanisms. While pharmacological approaches are often the first line of treatment, many patients fail to achieve adequate relief, leading to consideration of interventional strategies. Contemporary interventional options offer a wide array of techniques aimed at disrupting maladaptive pain pathways, each with distinct mechanisms, advantages and limitations [1,4,6].

Peripheral nerve interventions, including UGANL, RFA and CNL, address nociceptive input contributing to residual limb pain, with particular relevance for neuroma-related mechanisms. These techniques could provide effective, localized pain relief and are particularly advantageous due to their minimally invasive nature and relative procedural simplicity. However, evidence supporting their use primarily stems from small case series and lacks large-scale randomized controlled trials, limiting the generalizability of results [11,12,13].

Furthermore, traditional SCS has demonstrated meaningful pain reduction in a subset of patients with PAP, yet challenges remain regarding optimal paresthesia coverage [15,16]. Radiofrequency stimulation of the DRG offers a more targeted approach with better somatotopic specificity, while CES presents a novel alternative for patients with refractory symptoms not adequately addressed by conventional SCS [7,17].

PNS has recently gained prominence as a less invasive option capable of addressing both acute and chronic phases of PAP. PNS demonstrates favorable safety profiles, ease of placement and promising efficacy in reducing RLP and PAP [25,26].

However, each interventional modality carries its own limitations and potential complications. For instance, alcohol neurolysis, while simple and effective in many cases, may result in tissue damage or incomplete block of the ectopic impulses from the neuroma, requiring repeated procedures and/or more precise positioning of the needle tip [11]. CNL, although generally safe, can cause temporary skin changes, nerve injuries or unpredictable duration of action [13]. Neuromodulation techniques, particularly SCS, may be limited by lead migration, nerve root or spinal cord injury, wound infection or device-related complications [15,16]. The frequent absence of clear dermatomal pain patterns, driven by central nervous system reorganization postamputation, poses significant challenges in accurate DRG target selection, potentially compromising therapeutic efficacy [19]. CES lacks in experiences and data on reproducibility and safety across broader populations [7]. PNS, while promising, requires precise electrode placement and carries a risk of lead dislodgement, local infection and device malfunction. Furthermore, its long-term effectiveness and cost-efficiency in diverse clinical settings remain to be fully validated [24,26].

Overall, while contemporary interventional therapies expand the armamentarium for managing PAP, the current evidence base is still characterized by small sample sizes, heterogeneity in study designs and a predominance of observational studies. There is a critical need for high-quality randomized controlled trials comparing different interventional modalities head-to-head and for establishing standardized protocols for patient selection, procedural techniques and outcome assessment.

### 5.2. Limitations of Current Evidence

Despite the growing interest in interventional strategies for the management of PAP, the current body of evidence is characterized by several important limitations. Most available data derive from small case series, retrospective analyses and feasibility studies. While these studies provide valuable preliminary insights, they inherently carry risks of bias, limited generalizability and lack of standardized methodologies.

One of the major challenges in interpreting the existing literature is the considerable heterogeneity in patient populations, amputation levels, pain phenotypes (RLP versus PLP) and intervention techniques.

Additionally, outcome measures often vary widely between studies, ranging from pain intensity scales to functional assessments and quality of life metrics, complicating direct comparisons across interventions.

Another notable limitation is the scarcity of large, randomized controlled trials specifically comparing different interventional modalities head-to-head. Most available randomized controlled trials focus on isolated techniques, such as PNS or SCS, and lack direct comparisons to alternative approaches such as DRG stimulation or CNL. Consequently, there is insufficient evidence to establish clear treatment algorithms or to determine optimal patient selection for each modality.

Furthermore, the majority of studies are limited by relatively short follow-up durations, which restricts the ability to assess long-term efficacy, the durability of pain relief and the potential for delayed complications. This is particularly important in the context of neuromodulation techniques, where the sustainability of analgesic effects over years is a key consideration for clinical decision-making.

Finally, few studies adequately address cost-effectiveness, procedural risks or patient-centered outcomes such as satisfaction and functional recovery, all of which are critical elements in guiding evidence-based clinical practice.

### 5.3. Future Directions

Given the limitations of the current evidence base, future research in the management of PAP should prioritize the development of high-quality, methodologically rigorous studies. Large, multicenter randomized controlled trials are critically needed to directly compare different interventional modalities, such as SCS, DRG stimulation, CES, PNS and minimally invasive peripheral nerve interventions. Standardization of patient selection criteria, procedural protocols and outcome measures is essential to facilitate comparison across studies and to refine treatment algorithms. Stratifying patients based on pain phenotype (e.g., predominant RLP versus PLP), amputation level and underlying mechanisms of pain could help in optimizing therapeutic strategies and personalizing interventions.

Moreover, future studies should incorporate long-term follow-up periods to evaluate the durability of analgesic effects and identify late-onset complications. Given the potential neuroplastic changes associated with neuromodulation therapies, longitudinal studies assessing functional recovery, quality of life and patient satisfaction are warranted. Emerging areas of interest include the exploration of combinatory approaches, such as the concurrent use of pharmacological agents with interventional techniques and the application of neuromodulation earlier in the course of pain development to potentially prevent the transition to chronic pain.

Additionally, studies evaluating the cost-effectiveness and health economic impacts of different interventional strategies would be valuable for informing health care policy and resource allocation.

Finally, advances in imaging, neurophysiological monitoring and device technology may further refine targeting accuracy, improve efficacy and reduce procedural risks, paving the way for more effective and individualized treatments for patients suffering from PAP.

## 6. Conclusions

PAP represents a multifactorial and often debilitating condition that continues to pose significant therapeutic challenges. While contemporary interventional approaches offer a diverse array of minimally invasive techniques targeting both peripheral and central pain mechanisms, their broader clinical application remains constrained by limited high-quality comparative evidence, heterogeneity in clinical practices and relatively short follow-up periods. Nevertheless, these modalities hold considerable promise in selected patients when integrated thoughtfully into a comprehensive and individualized management strategy.

Moving forward, the development of standardized clinical algorithms and evidence-based treatment pathways is critical to guide optimal modality selection and sequencing. Future research efforts should prioritize robust comparative trials, long-term outcome evaluation and integration of patient-centered outcomes to better inform clinical decision-making. Ultimately, continued innovation and more rigorous evidence generation in this field are essential to advance pain management and enhance the quality of life for individuals living with PAP.

## Figures and Tables

**Table 1 biomedicines-13-01575-t001:** Peripheral nerve interventions.

Authors	Procedure	PLP/RLP/Both	Key Findings
Zhang et al. [11]	1–3 UGANL + 2 RFA *	Both	A 60% PLP reduction; 85% RLP reduction. A total of 54% responded to UGANL alone; others to RFA.
	13 subjects		
			(6 months **)
Pu et al. [12]	1–3 RFA	Both	A 69.2% PLP reduction and 82.4% RLP reduction.
	18 subjects		(12 months **)
Ilfeld et al. [14]	CNL vs. sham (placebo)	PLP	No significant difference: −0.5 vs. −0.0 *** improvement seen in selected subgroups.
	144 subjects		
			(4 months **)

* RFA was administered only in patients who did not achieve satisfactory pain relief following UGANL; ** follow up period; *** numeric rating scale 0–10 was used to measure pain.

**Table 2 biomedicines-13-01575-t002:** Neuromodulation strategies.

Modality	Study/Author(s)	Key Findings	PLP/PAP/Both
SCS	Corbett et al. [15]; Jaffee et al. [16]; Aiyer et al. [18]	Mixed outcomes; 7/12 studies reported benefit (Aiyer); 50% pain relief in select patients (Jaffee); limited by small, uncontrolled designs (Corbett)	PAP
DRG	Hunter et al. [19]; Eldabe et al. [20]	60–90% pain reduction (Hunter); mean pain reduction of 52%; 5/8 patients with significant relief (Eldabe); success linked to lead placement	PAP
CES	Lee et al. [7]	Case report: 80% pain reduction; good coverage for distal extremity pain; promising in refractory cases	PAP
PNS	Gilmore et al. [26]; Cohen et al. [27]	67% with ≥ 50% pain relief at 12 months (Gilmore); 75% with ≥ 50% relief at 60 days (Cohen); improved function and satisfaction	both

## Data Availability

All data are available in references.

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
