# Peer review of "Targeted Interventional Therapies for the Management of Postamputation Pain: A Comprehensive Review"

_biomedicines, 2025, doi:10.3390/biomedicines13071575_

Round 1
Reviewer 1 Report
Comments and Suggestions for Authors
Savicevic and collegues wrote a narrative review that presents a comprehensive and up-to-date synthesis of current interventional therapies for the treatment of postamputation pain (PAP), including both residual limb pain (RLP) and phantom limb pain (PLP). The authors describe the multifactorial pathophysiology of PAP and review a range of targeted interventional modalities including peripheral nerve interventions such as ultrasound-guided alcohol neurolysis (UGANL), radiofrequency ablation (RFA), and cryoneurolysis (CNL), as well as neuromodulation techniques including spinal cord stimulation (SCS), dorsal root ganglion (DRG) stimulation, cauda equina stimulation (CES), and peripheral nerve stimulation (PNS). The authors conclude that interventional strategies are promising but remain under-validated by large-scale clinical trials. The review calls for further standardization, outcome evaluation, and personalized treatment algorithms in the clinical management of PAP.
I recommend that the paper be accepted for publication. Below I list the major and minor issues that the authors should consider addressing before final acceptance.
Major
1)The authors state the review to be a narrative review but the approach resembles a systematic review. To improve clarity about the intended level of methodological rigor, the authors should consider explaining why this approach was chosen and provide a brief explanation for the lack of PRISMA adherence.
Minor
1)The authors mentioned promising outcomes for the different techniques, but no formal grading of evidence was provided. Comparing the level of evidence supporting each therapy would improve usability. For example adding a summarizing table with modality, mechanism, evidence source, reported efficacy and limitation for each technique would be useful for clinical readers and researchers.
2) The individual techniques are well reviewed, however providing direct comparisons, for example by adding a brief summary or paragraph that compares modalities head to head in terms of invasiveness, efficacy, cost, etc would strengthen the review.
3) Some studies are used to justify treatment recommendation, but are only briefly referenced without providing details about the methodology, effect size, etc. Providing more details and key statistics would inform the reader about the strength of evidence.
Author Response
Comment 1: Savicevic and collegues wrote a narrative review that presents a comprehensive and up-to-date synthesis of current interventional therapies for the treatment of postamputation pain (PAP), including both residual limb pain (RLP) and phantom limb pain (PLP). The authors describe the multifactorial pathophysiology of PAP and review a range of targeted interventional modalities including peripheral nerve interventions such as ultrasound-guided alcohol neurolysis (UGANL), radiofrequency ablation (RFA), and cryoneurolysis (CNL), as well as neuromodulation techniques including spinal cord stimulation (SCS), dorsal root ganglion (DRG) stimulation, cauda equina stimulation (CES), and peripheral nerve stimulation (PNS). The authors conclude that interventional strategies are promising but remain under-validated by large-scale clinical trials. The review calls for further standardization, outcome evaluation, and personalized treatment algorithms in the clinical management of PAP. I recommend that the paper be accepted for publication. Below I list the major and minor issues that the authors should consider addressing before final acceptance.
Answer 1: We thank the reviewer for his comments.
Comment 2: The authors state the review to be a narrative review but the approach resembles a systematic review - improve clarity.
Answer 2: We thank the reviewer for his comments, we have now further emphasized this methodological choice to avoid confusion and to clarify that, although the structure may resemble that of a systematic review, our intent was to offer a narrative synthesis.
Comment 3: Summarizing table with modality, mechanism, evidence source, reported efficacy and limitation for each technique would be useful for clinical readers and researchers.
Answer 3: Thank you for your comment. We have added two summary tables.
Comment 4: Providing direct comparisons, for example by adding a brief summary or paragraph that compares modalities head to head in terms of invasiveness, efficacy, cost, etc would strengthen the review.
Answer 4: We thank the reviewer for his comments. However, such direct comparisons were often limited by the heterogeneity and methodological variability among the included studies as we have addressed this issue in the Discussion and Limitations section. There, we emphasize the need for future comparative trials with clearly defined protocols and outcome measures.
Comment 5: Providing more details and key statistics would inform the reader about the strength of evidence.
Answer 5: Thank you for your comment. We have now added more details and key statstics.
Reviewer 2 Report
Comments and Suggestions for Authors
The presented literature review appears to be very simplistic in its considerations regarding the characteristics of PAP, with inaccuracies in the specific definitions of the complex pathology it is addressing. It shows limitations in the overview of the various therapeutic options available in the literature, a lack of an expository synthesis of the different options presented by the individual selected studies (it seems strange to me that there are only 21 eligible studies), and makes no mention of QOL (quality of life) issues within the context of the review. Paragraph 3, 'Pathophysiology of Postamputation Pain,' should be structured into several sections aimed at better explaining the components and mechanisms involved in nociceptive, nociplastic, and neuropathic pain. Furthermore, the epidemiology and prevalence are treated very superficially and imprecisely. There is a lack of a summary table highlighting the key points and the different categories of approaches found in the accepted studies.
Author Response
Comment 1: The presented literature review appears to be very simplistic in its considerations regarding the characteristics of PAP, with inaccuracies in the specific definitions of the complex pathology it is addressing.
Answer 1: We thank the reviewer for the comments. We have now revised the relevant section to provide a more detailed and structured explanation of the characteristics of postamputation pain.
Comment 2: It shows limitations in the overview of the various therapeutic options available in the literature,
Answer 2: Thank you for your answer. We have now added other therapeutic options.
Comment 3: Only 21 eligible studies, and makes no mention of QOL (quality of life) issues within the context of the review.
Answer 3: We thank the reviewer for the comments. We added more studies to strengthen the evidence base. Also, the issue of quality of life (QOL), previously omitted, is now integrated into the manuscript.
Comment 4: Paragraph 3, 'Pathophysiology of Postamputation Pain,' should be structured into several sections aimed at better explaining the components and mechanisms involved in nociceptive, nociplastic, and neuropathic pain.
Answer 4: We thank the reviewer for the comments. Paragraph 3 has been substantially revised and reorganized, as suggested.
Comment 5: The epidemiology and prevalence are treated very superficially and imprecisely.
Answer 5: Thank you for the comment. We now corrected this.
Comment 6: Lack of a summary table.
Answer 6: We appricate your comment. We have now added summary tables.
Round 2
Reviewer 2 Report
Comments and Suggestions for Authors
I've reviewed the authors' responses and the corrections made to the final text. It appears all the requested additions have been carried out.